## [Transparent Peer Review file · Nature Communications]

Proton channels govern vesicular carbonate chemistry in mineralizing cells of a marine calcifier

Corresponding Author: Dr Marian Hu

Version 0:

Reviewer comments:

Reviewer #1

(Remarks to the Author)

This paper describes experiments that claim to demonstrate that the Otop2l channel is involved with calcification of sea urchin cytoskeleton. The paper does show that Otop2l is expressed in the relevant cell type of sea urchin larvae, at both the plasma membrane and in vesicular membranes, and the authors make a reasonable case for the role of alkalinization of the vesicles during skeleton formation. However, the paper suffers from several major flaws which prevent it from demonstrating Otop2l relevance to this process.

The authors do not check for the presence of the Hv1 channel in PMC cells in question, even though Hv1 is known to be expressed in *S. purpuratus*. Many of the effects, including sensitivity to membrane potential (!) that they see could be explained by Hv1 action. The authors do not address this point at all.

Also experiments shown in Fig 3E lack any change in reversal potential with changing pHext. The reversal potential stays at approximately -80 mV over a >1.5 unit change in pHext. Other Otop papers (e.g., PMID: PMC9348849 as just one example)

show expected behavior of the Otop channel with regard to change in reversal potential with pH. This strongly indicates that the current measured in the heterologous system shown in Fig 3E here is not proton selective.

Although I had other questions, these points alone make it difficult to consider the rest of the paper and I recommend rejecting it for that reason.

Reviewer #2

(Remarks to the Author)

The manuscript by Jonusaite et al presents new evidence for the role of otopetrin H⁺ channels as key components of the transport processes driving calcification, using the sea urchin larva model system. The role of otopetrins in sea urchin spicule formation was proposed by this group in an earlier PNAS paper. The current manuscript takes the work significantly further and proposed roles for otopetrins in mediating H⁺ fluxes at both the plasma membrane and intracellular vesicle membranes. The importance of intracellular vesicles in mediating calcite precipitation is well-articulated and this work potentially provides a significant advance in understanding what are likely to be more general principles of calcification in a range of different organisms.

The manuscript presents a significant body of work using primarily fluorescent indicators to visualize cytosolic and vesicle pH, as well as cell membrane potential and calcium localization. It also presents new data obtained by heterologous expression in *Xenopus* oocytes and HEK cells. The manuscript is well written and generally easy to follow. The manuscript raises a several questions and requires further clarification on a number of points:

Major points

- Fig. 2B. The use of DIBAC4 to monitor membrane potential. It is noted (line 196) that “on some occasions” strong dye fluorescence was observed in sub-cellular vesicles and vacuoles. To what extent will this compartmentalization, some of which may not be visually apparent in the images, lead to errors in estimating plasma membrane (PM) potential? Clearly if the cells are able to actively accumulate dye in to sub-cellular compartments then the intracellular fluorescence will not

accurately reflect simply the PM potential.

- Fig. 3 reports effects of Otop21 knock down on vesicular H⁺ permeability. What effects did knock down have on cytosolic pH? It would seem important to have this information since changes in vesicular pH seem likely to be driven by changes in cytosolic pH. It is not possible to infer changes in vesicular H⁺ permeability without knowing the changes in cytosolic pH.

- Fig. 3 E. why is the reversal potential constant at all pH values? This would be expected to shift (maximally by 58 mV per pH unit) as the equilibrium potential changed, unless intracellular pH also shifted proportionally with external pH. This seems unlikely but could be addressed by monitoring intracellular pH in the oocytes. Another possibility is that the zero current values are not true due to incorrect subtraction of leak current. This should be checked.

- Figs 3G,I. Interestingly, unlike the pH-induced currents, the reversal potential does shift with increasing Mg²⁺ and Ca²⁺. The slight positive shift in reversal with increasing Mg²⁺ may indicate a permeability to Mg²⁺. However, the negative shift in reversal with increasing Ca²⁺ is hard to explain. Again, the zero current values should be checked.

- Fig. 4, lines 306-308. The increased cytosolic pH (to 8.35) during recalcification would potentially render the channel-mediated efflux of H⁺ from the cell ineffective, unless PM potential was depolarized accordingly. However, the authors state (lines 311-312) that membrane potential remained unchanged. This needs some discussion.

- Discussion, lines 374-377 and Fig. 5. It is not clear how channel-mediated H⁺ removal from the intracellular vesicles can occur when the vesicle pH is approaching 8.5. A vesicle membrane potential of at least -60 mV (cytosol negative) would be required to bring this about, assuming cytosolic pH of 7.5. Is it possible to estimate vesicle membrane potential from cells that had accumulated DiBAC4 into intracellular vesicles and vacuoles? The authors may also wish to discuss the possibility that channel-mediated H⁺ efflux from vesicles is more effective at lower vesicle pH so may be involved in alkalization of vesicles from lower pH values.

Minor points

- Fig 2F should be 2E. it is not clear how the membrane potential value was calculated from the arbitrary fluorescence values.

- Lines 491-497. An additional explanation for the different dye localizations could be the actual existence of alkaline vesicles that are not involved in calcium accumulation.

Version 1:

Reviewer comments:

Reviewer #2

(Remarks to the Author)

I have read carefully the revised manuscript and the authors' detailed responses to my earlier concerns and requests for clarification. I am satisfied the the authors have addressed these as much as possible and that the revised manuscript represents a significant improvement. I have no further comments other than to say that this is a substantial and novel study.

REVIEWER COMMENTS

Reviewer #1 (Remarks to the Author):

This paper describes experiments that claim to demonstrate that the Otop2l channel is involved with calcification of sea urchin cytoskeleton. The paper does show that Otop2l is expressed in the relevant cell type of sea urchin larvae, at both the plasma membrane and in vesicular membranes, and the authors make a reasonable case for the role of alkalization of the vesicles during skeleton formation. However, the paper suffers from several major flaws which prevent it from demonstrating Otop2l relevance to this process.

Author's reply:

We thank reviewer#1 for seeing the overall merit of this work in revealing the biological role of Otop2l in the cellular mineralization process, but also the strong but constructive criticism on some major issues presented in this manuscript. Reviewer#1 raised valid points that we adequately addressed by adding new experiments and analyses which clearly demonstrate that the sea urchin otop2l encodes a proton selective channel in the calcifying cells (PMCs) of the sea urchin larva. In addition, we provide new transcriptomic evidence that Otop2l is the major proton channel expressed by PMCs.

This work builds on a previous study by Chang et al. 2021 PNAS 118 (30) e2101378118 where we characterized the sea urchin Otop2l for the first time. In line with the present work, we see a clear shift in the reversal potential (E_{rev}) in the pH range from 7.4-8.5 that follows the Nernst potential for protons, which is hard to see in figure 3E of the present manuscript. We added new analyses and experiments clarifying the observed shifts in E_{rev} using both *Xenopus* oocytes and HEK293 cells expressing Otop2l. Thus, we are very confident that Otop2l encodes a proton selective channel. Please refer to our specific comments for more details regarding this important issue.

We would also like to mention that besides the electrophysiological characterization of the Otop2l proton channel, the present work provides a substantial amount of new information that is of fundamental importance for the cellular physiology of biomineral formation in animals. This includes *in vivo*-tracking of pH and Ca^{2+} ions in living cells where we for the first time observe the process of mineral deposition in an animal. Furthermore, to our best knowledge this is the first study investigating carbonate system variables in the calcification vesicle and how these conditions are affected by environmental pH changes.

We feel that we adequately addressed the major concerns raised by reviewer#1 and hope that our revisions are acknowledged along with the merits of this study, which we believe contributes significantly to the field of biomineralization and cell-physiology.

Reviewer#1

The authors do not check for the presence of the Hv1 channel in PMC cells in question, even though Hv1 is known to be expressed in *S. purpuratus*. Many of the effects, including sensitivity to membrane potential (!) that they see could be explained by Hv1 action. The authors do not address this point at all.

Author's reply:

We thank reviewer#1 for this valid criticism of our manuscript. We indeed found the gene encoding Hvcn1 (VSOP) in the genome of the purple sea urchin. However, the expression levels of this gene are very low compared to Otop2 (echinobase.org). We further used our single cell transcriptome to check for the tissue specific expression of Hvcn1 and Otop2l in the sea urchin larva and confirmed almost absent expression of Hvcn1 in any larval tissue along the early development (24-72 h post fertilization), which is the critical time window for skeleton formation. In contrast, Otop2l is highly and specifically expressed in the calcifying primary mesenchyme cells across all relevant developmental stages. We added a new figure S7 to our supplemental material presenting the

single cell transcriptomic analyses and included this important point in our results (L248-252) and discussion (L381-383) section. Thus, we are very confident that Otop2l is the major player in the herein observed proton permeability of PMCs.

Reviewer#1

Also experiments shown in Fig 3E lack any change in reversal potential with changing pH_{ext}. The reversal potential stays at approximately -80 mV over a >1.5 unit change in pH_{ext}. Other Otop papers (e.g., PMID: PMC9348849 as just one example) show expected behavior of the Otop channel with regard to change in reversal potential with pH. This strongly indicates that the current measured in the heterologous system shown in Fig 3E here is not proton selective.

Author's reply:

We thank reviewer#1 for pointing out this important concern. However, we are very confident that the sea urchin Otop2l encodes a proton selective channel based on several evidences. In our previous paper by Chang et al. 2021 PNAS 118:e2101378118, we demonstrate that in the pH range from 7.5 to 8.5 the reversal potential shifts according to the Nernst potential for protons. In the same study we demonstrated using intracellular pH measurements in combination with external pH changes that proton conductance in both HEK293 cells and primary mesenchyme cells of the sea urchin are strongly increased when expressing Otop2l.

We agree that the voltage ramps shown in Fig. 3E are suggesting that there is no shift in E_{rev} with changing pH which is due to two reasons. First, the thick lines make it hard to see that there is a clear shift in E_{rev} in the pH range from 7.4 to 8.5. We have now added the inset to revised Fig. 3E showing an enlarged view indicating the shift in reversal potential in the pH range from 7.4 to 8.6. We performed new experiments and added three new supplemental figures (fig S9 A,B,D) demonstrating this shift in E_{rev} in the physiologically relevant pH range from pH 7.4 to 8.6 by 50-58 mV using both *Xenopus* oocytes and HEK293 cells expressing Otop2l. Second, indeed in our oocyte experiments the shift in E_{rev} decreased at pH levels exceeding > pH 8.6. This point was also raised by the second reviewer, and we followed his/her suggestion to measure pH_i using pH selective microelectrodes. Interestingly, we observed an increase in pH_i at higher external pH conditions, explaining the reduction in the shift in E_{rev} due to unchanged or even diminishing proton gradients at these highly alkaline pH_e conditions. The results are presented in table S5 and from these pH changes we calculated the theoretical reversal potential that is additionally plotted in Fig. S9 A+B). Taken together, we are very confident about our conclusions that the herein measured currents for Otop2l in *Xenopus* oocytes and HEK293 cells are selective to protons and importantly that the channel is activated by alkaline pH. We added this important point to our results section (Lines 262 – 267).

Reviewer#1

Although I had other questions, these points alone make it difficult to consider the rest of the paper and I recommend rejecting it for that reason.

Author's reply:

We understand the criticism of reviewer#1 and adequately addressed all points raised to demonstrate that our measurements and conclusions presented in this work are valid. We hope that our additional experiments and explanations convince reviewer#1 and that he/she sees the merit of this novel and comprehensive study on pH regulatory strategies in mineralizing cells.

Reviewer #2 (Remarks to the Author):

The manuscript by Jonusaite et al presents new evidence for the role of otopetrin H⁺ channels as key components of the transport processes driving calcification, using the sea urchin larva model system. The role of otopetrins in sea urchin spicule formation was proposed by this group in an earlier PNAS paper. The current manuscript takes the work significantly further and proposed roles

for otopetrins in mediating H⁺ fluxes at both the plasma membrane and intracellular vesicle membranes. The importance of intracellular vesicles in mediating calcite precipitation is well-articulated and this work potentially provides a significant advance in understanding what are likely to be more general principles of calcification in a range of different organisms.

The manuscript presents a significant body of work using primarily fluorescent indicators to visualize cytosolic and vesicle pH, as well as cell membrane potential and calcium localization. It also presents new data obtained by heterologous expression in *Xenopus* oocytes and HEK cells. The manuscript is well written and generally easy to follow. The manuscript raises a several questions and requires further clarification on a number of points:

Author's reply:

We thank reviewer#2 for the positive and constructive response to our work. In our revised manuscript we addressed all major issues and added new experiments that significantly improved the quality of this manuscript. In particular, in our oocyte and HEK293 cell experiments we did not clearly explain that we measured a shift in the reversal potential with changes in external pH following the Nernst potential for protons in the pH range of 7.4-8.6 but at higher pH values the reversal potential fails to show the expected shift. Here we followed reviewer's suggestion and measured intracellular pH of oocytes using pH microelectrodes and observed that indeed pH_i starts to increase at alkaline external conditions >pH 8.6. We modified figure 3 and added three new figures (S9) to our supplemental material to clarify this point. Furthermore, we addressed the important concern regarding the proton gradients across the plasma and vesicular membranes to promote proton export from the calcification vesicle and the cytosol. Here we added new experiments determining cytosolic pH changes in control and Otop2l knock-down larvae during external pH changes. Finally, we used another membrane potential dye to estimate the vesicular membrane potential in relation to the plasma membrane potential, suggesting a high lumen positive potential that could drive efflux of protons from the vesicle. We carefully addressed all points in point-by-point manner and revised our manuscript accordingly, and feel that these changes significantly improved the quality of this manuscript.

Major points

Reviewer#2

- Fig. 2B. The use of DIBAC4 to monitor membrane potential. It is noted (line 196) that "on some occasions" strong dye fluorescence was observed in sub-cellular vesicles and vacuoles. To what extent will this compartmentalization, some of which may not be visually apparent in the images, lead to errors in estimating plasma membrane (PM) potential? Clearly if the cells are able to actively accumulate dye in to sub-cellular compartments then the intracellular fluorescence will not accurately reflect simply the PM potential.

Author's reply:

The primary goal of these experiments was to monitor relative changes in the plasma membrane potential caused by alterations in external [K⁺] and other ions and their effect on pH_i and pH_v. For this purpose, a potential background fluorescence from vesicles with strong DIBAC fluorescence would be uncritical. The determination of the apparent membrane potential (V_{app}) using DIBAC4 (we now added a detailed description for this calculation to our revised manuscript) was used to obtain an alternative measurement for the resting membrane potential in PMCs in addition to previously published measurements using microelectrodes (Chang et al. 2011 PNAS). Here we agree that for the determination of the apparent membrane potential (V_{app}) based on the fluorescence intensities within the cytosol and the extracellular medium, accumulation of the dye in vesicles could indeed cause artifacts. However, based on several observations we are confident that our determination of V_{app} is relatively accurate. First, the V_{app} obtained from our DIBAC4 measurements (-26 ± 6.9 mV) is very close to our direct electrode measurements of -20.6 mV. For the measurements in the present study, we only determined V_{app} in cells that showed a uniform fluorescence throughout the cytoplasm (no vesicular staining). In addition, the pH change of the cytosol by ca. 0.5 pH units would require a depolarization by 29.6 mV which is again very close to the actual measured change in V_{mem} (26 mV).

Finally, to our best knowledge DiBAC₄(3) emits fluorescence in contact with cytosolic or membrane proteins (i.e. <https://www.thermofisher.com/order/catalog/product/de/en/B438>) and thus strong fluorescence of vesicles should not reflect accumulation within vesicles but potentially strong potential differences across the vesicle membranes. We understand the reviewers' concern and hope that our explanation clarifies it, and that we were able to convince reviewer#2 that the results of membrane potential and pH changes justify the conclusions drawn in the manuscript.

Reviewer#2

- Fig. 3 reports effects of Otop21 knock down on vesicular H⁺ permeability. What effects did knock down have on cytosolic pH? It would seem important to have this information since changes in vesicular pH seem likely to be driven by changes in cytosolic pH. It is not possible to infer changes in vesicular H⁺ permeability without knowing the changes in cytosolic pH.

Author's reply:

We agree to this comment by reviewer#2 and added new p*H*_i measurements in control and Otop21 knock-down larvae to be able to reconstruct all pH gradients from the environment over the cytoplasm to the vesicular compartments. We used the same experimental protocol as for the vesicular pH measurements of this study, and found that knock-down cells have attenuated pH changes (new figure S8) similar to those measured in our previous study by Chang et al. 2021. We included these new results in our revised manuscript and discussed how attenuated proton permeability may affect cytosolic and vesicular pH.

Reviewer#2

- Fig. 3 E. why is the reversal potential constant at all pH values? This would be expected to shift (maximally by 58 mV per pH unit) as the equilibrium potential changed, unless intracellular pH also shifted proportionally with external pH. This seems unlikely but could be addressed by monitoring intracellular pH in the oocytes. Another possibility is that the zero current values are not true due to incorrect subtraction of leak current. This should be checked.

Author's reply:

We fully understand this concern raised by reviewer#2 and acknowledge the confusing representation of the measurements in figure 3E. In our previous paper by Chang et al. 2021 PNAS 118:e2101378118, we demonstrate that in the pH range from 7.5 to 8.5 the reversal potential shifts according to the Nernst potential for protons. This is also true in the present study (e.g. a shift by 50-58 mV per decade change in H⁺ conc.), and we added an additional figure (S9) to demonstrate the shift in E_{rev} according to the Nernst potential for protons in this pH range when using both *Xenopus* oocytes with different p*H*_e and a p*H*_e change from pH7.4 to 8.5 in HEK293 cells. However, in our oocyte experiments the shift in E_{rev} decreased at pH levels exceeding > pH 8.6. Thus, we followed reviewer's suggestion to measure p*H*_i using pH selective microelectrodes. Interestingly, we observed an increase in p*H*_i at higher external pH conditions, explaining the reduction in the shift in E_{rev} due to unchanged or even diminishing proton gradients at these highly alkaline pH conditions. We added these results and a table of these measurements to our supplemental material. Taken together, we are very confident concluding that our measurements in *Xenopus* oocytes and HEK293 cells demonstrate proton selective currents of Otop21.

Reviewer#2

- Figs 3G,I. Interestingly, unlike the pH-induced currents, the reversal potential does shift with increasing Mg²⁺ and Ca²⁺. The slight positive shift in reversal with increasing Mg²⁺ may indicate a permeability to Mg²⁺. However, the negative shift in reversal with increasing Ca²⁺ is hard to explain. Again, the zero current values should be checked.

Author's reply:

We think that the negative shift in reversal potential in our Ca²⁺ experiments is based on a misunderstanding of our experimental protocol. The grey line is a recording at pH 7.4 and all recordings for different Ca²⁺ concentrations were performed under pH 8.5 activation of the channel. This is why there is a strong negative shift between the pH 7.4 trace and all other Ca²⁺ concentrations. Please refer to figure 3J for a clarification of the experimental protocol. In addition, we have added a new figure (Fig. S9C) to our supplemental material that demonstrates a shift in reversal potential following the Nernst potential for protons from pH 7.4 to 8.5 and no further change in reversal potential at various Ca²⁺ concentrations.

In the original traces shown in Fig 3G, we cannot see a positive shift with increasing Mg²⁺ concentrations, but if at all a very slight negative shift. We carefully checked our Mg²⁺ measurements, and cannot find a slight positive shift in reversal potential with increasing Mg²⁺ as pointed out by reviewer#2. We think that the slight negative shift may be caused by an activation of the channel by Mg²⁺ and due to small proton gradients across the plasma membrane.

Reviewer#2

- Fig. 4, lines 306-308. The increased cytosolic pH (to 8.35) during recalcification would potentially render the channel-mediated efflux of H⁺ from the cell ineffective, unless PM potential was depolarized accordingly. However, the authors state (lines 311-312) that membrane potential remained unchanged. This needs some discussion.

Author's reply:

This question raised by reviewer#2 is well taken and we agree that under these pH conditions proton efflux from the cell should stop/reverse. In previous studies, we determined an intracellular pH in PMCs ranging between 6.8 to 7.2 (Stumpff et al. 2012 PNAS 109 (44) 18192-18197; Hu et al 2018 elife <https://doi.org/10.7554/eLife.36600>, Chang et al. 2021 PNAS 118 (30) e2101378118, Matt et al. PNAS 119 (40) e2203904119). We also previously observed an increase in pH_i during re-mineralization of the skeleton by approximately 0.3 pH units (Hu et al 2020 Proc. Roy. Soc B <https://doi.org/10.1098/rspb.2020.1506>). Therefore, we carefully re-evaluated pH_i measurements in the current study and found an error in the transformation of the sigmoidal fit to calculate pH values from ratio values. We now determined a pH_i of 6.84 for control and 7.02 for re-mineralizing PMCs, respectively. We apologize for this error and revised our manuscript accordingly. Under these circumstances even at a slightly (i.e. -20 to -25 mV) negative membrane potential protons would still exit the cell. At a V_M of -25 mV proton efflux would take place up to a pH_i of ca. 7.7 at external seawater conditions of pH 8.1. We added a short discussion regarding this point to our revised manuscript (I546-I548).

Reviewer#2

- Discussion, lines 374-377 and Fig. 5. It is not clear how channel-mediated H⁺ removal from the intracellular vesicles can occur when the vesicle pH is approaching 8.5. A vesicle membrane potential of at least -60 mV (cytosol negative) would be required to bring this about, assuming cytosolic pH of 7.5. Is it possible to estimate vesicle membrane potential from cells that had accumulated DiBAC4 into intracellular vesicles and vacuoles? The authors may also wish to discuss the possibility that channel-mediated H⁺ efflux from vesicles is more effective at lower vesicle pH so may be involved in alkalinization of vesicles from lower pH values.

Author's reply:

We agree that this is an intriguing question and central to the topic of vesicular pH regulation via a proton channel. As reviewer#2 suggested, we were tempted to speculate that due to high DiBAC4 fluorescence occasionally observed in PMCs there might be a strong potential difference across the vesicular membrane. However, we do not feel confident to draw this conclusion based on our DiBAC4 measurements due to the fact that DiBAC4 is an anionic dye that can freely diffuse across biological

membranes, and only emits fluorescence in the presence of proteins (within the cytoplasm). Thus strong fluorescence associated with vesicles would suggest a more positive (lumen negative) potential across the vesicle membrane. However, since we feel that this question is of central importance, we tried to use another membrane potential dye Di-8-ANEPPS which is only fluorescent in membranes and responds to hyperpolarization of the plasma membrane by an increase in fluorescence. Using this dye, we observed a strong vesicular signal in PMCs largely colocalizing with calcein which suggests endocytotic origin of the vesicles. The fact that this dye only enters the outer leaflet of the lipid bilayer of the plasma membrane (Biswas et al. 2024 L Mat Chem. B DOI:10.1039/D4TB00872C) further infers that this dye can only get into the vesicle membranes by endocytosis. In addition, strong fluorescence in these vesicles compared to fluorescence in the plasma membrane may point towards a stronger hyperpolarization of the vesicle membrane compared to the plasma membrane. We added a supplemental figure S6 with these results and integrated this finding in our discussion.

L551-565: "Even under elevated pH_i , the outwardly directed proton gradient and a membrane potential of -25mV would still promoteUsing Di-8-Anepps, a membrane potential probe that emits fluorescence when integrated into the outer leaflet of the plasma membraneexplain the electrochemical driving-forces for luminal alkalization in endocytotic vesicles."

Minor points

Reviewer#2

- Fig 2F should be 2E. it is not clear how the membrane potential value was calculated from the arbitrary fluorescence values.

Author's reply:

Thank you. We corrected the figure labels in the text. We also added a description to the methods section how V_{app} was calculated from the arbitrary fluorescence values using the Nernst equation.

Reviewer#2

- Lines 491-497. An additional explanation for the different dye localizations could be the actual existence of alkaline vesicles that are not involved in calcium accumulation.

Author's reply:

Yes, we agree, but it would be hard to explain how seawater endocytosis can selectively exclude calcein, which would show fluorescence in the presence of Ca^{2+} ions at concentrations found in seawater. Therefore, we suggested an active export of Ca^{2+} ions from some of the alkaline vesicles which would decrease or eliminate calcein fluorescence.

REVIEWERS' COMMENTS

Reviewer #2 (Remarks to the Author):

I have read carefully the revised manuscript and the authors' detailed responses to my earlier concerns and requests for clarification. I am satisfied the the authors have addressed these as much as possible and that the revised manuscript represents a significant improvement. I have no further comments other than to say that this is a substantial and novel study.

Author's Reply:

We are grateful for the positive feedback and the constructive comments in the previous review round made by the reviewer's and look forward to find this work published soon.